# The daytime feeding frequency affects appetite-regulating hormones, amino acids, physical activity, and respiratory quotient, but not energy expenditure, in adult cats fed regimens for 21 days

Alexandra Camara[1], Adronie Verbrugghe[1], Cara Cargo-Froom[2], Kylie Hogan[2], Trevor J. DeVries[2], Andrea Sanchez[1], Lindsay E. Robinson[3], Anna K. Shoveller[2]*

1 Department of Clinical Studies, Ontario Veterinary College, University of Guelph, Guelph, Ontario, Canada,
2 Centre for Nutrition Modelling, Department of Animal Biosciences, Ontario Agricultural College, University of Guelph, Guelph, Ontario, Canada, 3 Department of Human Health and Nutritional Sciences, College of Biological Sciences, University of Guelph, Guelph, Ontario, Canada

* ashovell@uoguelph.ca

**Data Availability Statement:** All data files are available from the Scholars Portal Dataverse for the

## Abstract

The effects of feeding frequency on postprandial response of circulating appetite-regulating hormones, insulin, glucose and amino acids, and on physical activity, energy expenditure, and respiratory quotient were studied in healthy adult cats. Two experiments were designed as a 2 x 3 replicated incomplete Latin square design. Eight cats, with an average body weight (BW) of 4.34 kg ± 0.04 and body condition score (BCS) of 5.4 ± 1.4 (9 point scale), were fed isocaloric amounts of a commercial adult maintenance canned cat food either once (0800 h) or four times daily (0800 h, 1130 h, 1500 h, 1830 h). Study 1 consisted of three 21-d periods. On day 14, two fasted and 11 postprandial blood samples were collected over 24 hours to measure plasma concentrations of ghrelin, GLP-1, GIP, leptin, PYY, insulin and amino acids, and whole blood glucose. Physical activity was monitored from day 15 to 21 of each period. In Study 2 indirect calorimetry was performed on the last day of each period. Body weight was measured weekly and feed intake recorded daily in both experiments. No effect of feeding regimen on BW was detected. Cats eating four times daily had lesser plasma concentrations of GIP and GLP-1 (P<0.05) and tended to have lesser plasma PYY concentrations (P<0.1). Plasma leptin and whole blood glucose concentrations did not differ between regimens (P>0.1). Cats fed once daily had a greater postprandial plasma amino acid response, and greater plasma ghrelin and insulin concentrations (P<0.05). Physical activity was greater in cats fed four times (P<0.05), though energy expenditure was similar between treatments at fasting and in postprandial phases. Finally, cats eating one meal had a lower fasting respiratory quotient (P<0.05). Overall, these data indicate that feeding once a day may be a beneficial feeding management strategy for indoor cats to promote satiation and lean body mass.

University of Guelph (doi.org/10.5683/SP2/PRDKEE).

**Funding:** AV and AKS received the grant Grant #W18-039 WINN Feline Foundation https://www.winnfelinefoundation.org/. The funders had no role in the study design, data collection and analysis, decision to publish, or preparation of the manuscript.

**Competing interests:** The authors declare no conflicts of interest. A.V. is the Royal Canin Veterinary Diets Endowed Chair in Canine and Feline Clinical Nutrition at the Ontario Veterinary College. This does not alter our adherence to PLOS ONE policies on sharing data and materials.

## Introduction

Research in feline nutrition has largely focused on the effects of nutrient distribution and density on feline physiology, while a derth of work has been paid to feeding management, such as feeding frequency. A recent consensus statement, which focused on reducing behavioural problems and stress in cats, recommended that cats' daily food allowances be divided into multiple small meals fed throughout the day [1]. There is, however, a lack of empirical evidence to support multiple meal feeding for the indoor pet cat to help support a healthy body composition and body weight (BW).

Numerous factors affect energy intake and utilization and excess intake creates positive energy balance and deposition of adipose tissue [2–4]. Obesity, which largely affects middle-aged cats [5–9], is the most common nutritional problem in cats and is considered to be a low-grade inflammatory disease [2–4], predisposing cats to many adverse health conditions [7, 10, 11]. A*d libitum* feeding and feeding frequently, which cannot be teased apart in epidemiological research, have been reported to be risk factors for weight gain and adverse health conditions by some [12–14], but not by others [5, 6, 8, 15, 16]. Courcier et al. [9] reported that cats fed twice daily were more likely to be obese than cats fed *ad libitum* [17]. In a recent review investigating meal frequency in human and animal (mouse, rat, calf, cow, horse, and fish) studies, it was suggested that increased eating frequency may not assist in reducing energy intake or improving BW [18]. However, cats had lesser plasma concentrations of orexigenic ghrelin, and greater concentrations of anorexigenic leptin when fed four times daily compared to twice daily [19]. Cats fed two or four times daily also had greater voluntary physical activity than cats fed once per day [20, 21]. LeBlanc and Diamond reported that increased meal frequency was related to greater postprandial thermogenesis [22] and this was likely related to increased anticipation of feeding [22]. Though energy expenditure (EE) was not affected by intermittent fasting in humans, the observed decrease in respiratory quotient (RQ) is suggestive of enhanced fat oxidation [23, 24]. To the authors' knowledge, research examining the effects of feeding frequency on energetics and macronutrient utilization have not been performed.

Meal frequency may also affect protein metabolism and lean body mass [LBM]. In neonatal animal models, including rats and piglets, intermittent fasting has resulted in greater protein synthesis [25–31]. Gazzaneo et al. [32] suggested that compared to continuous feeding, intermittent feeding enhances lean tissue accumulation by promoting a greater and more rapid postprandial amino acid (AA) and insulin response, initiating protein synthesis. Similarly, postprandial AA concentrations increased more when resistance-trained men ate less frequently throughout the day [33]. Geriatric cats are prone to loss of LBM and BW as a result of age-related changes of energy metabolism [34] and could benefit from feeding management practices that increase lean tissue maintenance. Lean body mass loss, sarcopenia, is a natural process of aging, which occurs in the absence of clinical disease [35]. By age 15, cats may lose one-third of the LBM that was present between the ages of 1 to 7 years [36]. Increasing protein synthesis and LBM by changing the feeding regimen of older cats could help mitigate the incidence of sarcopenia and improve overall functionality.

The objectives of this research were to investigate the effects of feeding frequency, one compared to four meals per day, on fasting and postprandial serum concentrations of appetite-regulating hormones (glucagon-like protein-1 (GLP-1), gastric inhibitory protein (GIP), ghrelin, leptin and peptide YY (PYY), glucose, insulin, AA, fasting and postprandial EE and RQ, and voluntary physical activity in domestic cats.

## Materials and methods

All procedures were reviewed and approved by the University of Guelph Animal Care and Use Committee (AUP#3881).

## Animals

Eight healthy adult neutered/spayed cats (four males and four females) of similar age (1–4 years) with an average BW of 4.34 ± 0.04 kg and BCS of 5.4 ± 1.4 were used. All cats were previously acclimated to Actical® activity monitors (Mini Mitter, Bend, OR), respiration chambers [37] and associated environments.

## Housing

Cats were housed together indoors in a free-living environment in the Department of Animal Biosciences at the University of Guelph. The room was environmentally controlled with a 12 h light– 12 h dark cycle, with the lights turning on at 0800 h and off at 2000 h. The room temperature was maintained at 20.0˚C and relative humidity was kept between 40–60%. The room surfaces were cleaned daily. Water was provided *ad libitum*. All cats had access to a variety of environmental enrichment sources such as toys, scratching posts, hide boxes, perches, beds, and climbing apparatuses. There were 10 litter boxes available and they were cleaned out twice daily. All cats received human interaction with familiar people for two hours per day, five days a week. Interaction included brushing, petting, voluntary play, and general room upkeep.

Four respiration calorimetry chambers (Qubit Systems; Kingston, ON, Canada) were made of Plexiglas and measured 146cm x 60cm x 89cm. Chamber size was reduced to 64cm x 60cm x 52cm using wooden frames, covered in polyethylene plastic to remove dead space and $CO_2$ recovery measured and confirmed. Each chamber contained a water bowl, litter box, a resting box, and a free area with a blanket and toy. Chambers, water and food bowls, litter boxes, toys, and blankets were cleaned daily.

## Diet

Cats were fed a commercially available canned cat food, formulated for adult maintenance. This chicken and liver-based food was formulated to meet or exceed AAFCO standards [38]. Proximate analysis was performed using AOAC methods for dry matter [39], crude protein [40], acid-hydrolysed fat [41], ash [42], minerals [43] and AOCS method for crude fiber [44] (Table 1). Food allowance was based on historical feeding records and metabolizable energy for individual cats that resulted in stable BW. Cats were individually fed in cages or respiration chambers at either 0800 h (once a day feeding) or 0800 h, 1130 h, 1500 h, 1830 h (four times a day feeding). Cats were permitted 90 min once per day or 20 min four times per day to consume their food, respectively, and all remaining food was removed, weighed, and orts recorded

**Table 1. Proximate analyses of the canned diet\* used in a feeding frequency trial.**

| Typical Analysis (% as fed) | |
| --- | --- |
| Moisture | 77.00 |
| Dry Matter | 23.00 |
| Protein | 11.90 |
| Fat | 7.27 |
| Crude fibre | 0.32 |
| Ash | 1.65 |
| NFE[a] | 1.86 |

[a]NFE = nitrogen-free extract = 100−(crude protein + crude fat + crude fibre + moisture + ash); [45].

\* Ingredients: chicken, animal liver, meat by-products, poultry by-products, natural and artificial flavors, salt, tricalcium phosphate, guar gum, potassium chloride, dl-methionine, vitamins, minerals, choline chloride, added color, taurine, water sufficient for processing.

in grams. While all cats were fed in separate cages, but in the same room at 0800 h, cats receiving multiple meals were moved to a different, yet familiar, room and fed separately at all other remaining times to ensure cats fed once per day could not observe the other cats being fed an additional 3 times per day.

## Experimental design

Meal frequency was tested in two separate studies, each using a 2 x 3 replicated incomplete Latin square design with two treatments (one vs. four times feeding frequency) and three periods. The third period allowed us to account for carry over effects of feeding frequency in such that we sought to understand whether a longer time receiving a feeding frequency altered our physiological response criteria. Because all cats repeated one of two treatments, this resulted in 12 experimental periods per treatment as four of eight cats repeated one of two treatments. For each of the three periods in each study, cats were assigned to one of two treatments; fed once a day (0800 h) or fed four times a day (0800 h, 1130 h, 1500 h, 1830 h). Each period had four groups of two cats (one male, one female) receiving one of two treatments. In Study 1, each period was 21 d and was separated by a 6-d transition period during which the cats were fed two times daily. Cats were fed twice a day in the transition period to avoid any longer-term adaptation to either of the feeding treatments. In Study 1, pre-prandial and postprandial plasma appetite-regulating hormone (GLP-1, GIP, leptin, ghrelin, PYY), insulin, amino acids and whole blood glucose, and voluntary physical activity were assessed on d 14 of each treatment period. In Study 2, indirect calorimetry was performed on the last day of each of the three 14-d treatment periods. Treatment design remained the same for both Study 1 and Study 2. Feed intake was measured on a daily basis throughout both studies, and BW and BCS were measured weekly after an overnight fast.

## Blood collection

On d 12 of each period in Study 1, cats were sedated with an intramuscular (IM) injection of butorphanol (Zoetis Canada Inc., Kirkland, QC, Canada) (0.3 mg/kg) and dexmedetomidine (Zoetis Canada Inc., Kirkland, QC, Canada) (0.02 mg/kg) mixed in the same syringe. Twenty minutes after injection, a 22 G IV catheter was placed in the cephalic vein (Becton Dickinson Canada Inc., Mississauga, ON, Canada). Propofol (Fresenius Kabi Canada Ltd, Richmond Hill, ON, Canada) was administered intravenously to effect (average total dose of 1.9 mg/kg) to facilitate central catheter placement. The area over one of the jugular veins was aseptically prepared and a 16 G x 10 cm jugular catheter (MILA International Inc., Florence, KY, USA) was inserted, secured, and primed with 0.5 ml of heparinized saline solution (10 IU/ml) to maintain patency. After catheter placement, the reversal agent atipamezole (Zoetis Canada Inc., Kirkland, QC, Canada) (0.02 mg/kg) was administered IM and a 24 hours washout period was allowed before blood sample collection to avoid any drug interference with the analytical technique. Cats were monitored constantly, during sedation, catheter placement, after administration of the reversal agent, and throughout the entire period that the catheters were in place. Catheter patency was maintained by injecting 0.3 ml heparinized saline solution (1 IU/ml) after every blood sample (first eleven collection points), and 0.5 ml heparinized saline solution (10 IU/ml) after the 12th sample to avoid clotting overnight.

On d 14 of each treatment period, two fasted baseline blood samples (30 min and 5 min before the first meal) and 11 postprandial blood samples were taken at 60, 120, 180, 240, 360, 480, 600, 720, 900, 1080, and 1440 minutes after the first meal (0800 h). Blood samples (1.5 ml) were collected to measure plasma concentrations of appetite-regulating hormones (GLP-1, GIP, PYY, ghrelin and leptin), insulin, AA, and whole blood glucose concentrations. BD

Vacutainer tubes (2.0 mL, lavender top, Vacutainer™, Becton Dickinson, Franklin Lakes, NJ, USA) coated with EDTA containing the Dipeptidyl Peptidase-IV (DPP-IV) inhibitor (Millipore Sigma, Billerica, MA, USA), protease inhibitor (Sigma-Aldrich, St. Louis, MO, USA), and Pefabloc SC inhibitor (Sigma-Aldrich, St. Louis, MO, USA) were used to ensure the blood did not clot and before centrifugation and to prevent degradation of hormones. Whole blood glucose was assessed in duplicate directly after collection using a portable glucose meter, previously validated for use in cats (AlphaTRAK 2®, Abbott Laboratories, North Chicago, IL) [46, 47]. Samples were placed on ice and centrifuged (Beckman Coulter, J6-MI, Mississauga, ON, Canada) within 30 min of sampling at 1000 x g for 10 min and plasma was aliquoted and stored at -20˚C until further analyses. Duplicate analyses were performed for active GLP-1, GIP, PYY, ghrelin, leptin and insulin using a commercial magnetic bead test kit (Milliplex® MAP feline metabolic hormone magnetic bead panel, EMD Millipore Corporation, Billerica, MA, USA) following manufacturer's protocol and run on a Bio-Plex® 2000 system/ Bio-Plex Manager Software, version 6.0 (Bio-Rad, Canada). The quality control samples and standard curves were evaluated based on manufacturer recommendations. Coefficients of variation (CV) for the results were assessed for each set of duplicates. If CV per duplicate was <20%, the results were deemed acceptable and the average of the duplicates was used for further analysis. If the CV was >20%, individual results were assessed, and if duplicates were not in agreement, the results from that sample were removed.

AA standards and plasma (10 μL) were derivatized by ACCQTag Ultra derivatization kit (Waters Corporation, Milford, MA, USA) according to Boogers et al. [48]. The derivatized AA were separated using UPLC with UV detection (260 nm) with a cycle time of 10 min per sample. Derivatized AA (1 μL injection volume) were separated in a column (2.1 x 2000 mm, 1.7 μL) maintained at 55˚C. AA peak areas were compared with known standards and analyzed using Waters Empower 2 Software (Waters Corporation, Milford, MA, USA).

Homeostatic model assessment of insulin resistance (HOMA-IR) and glucose: insulin were calculated using the following equations [49]:

$$\text{HOMA}-\text{IR} = (\text{Insulin (uIU/mL)} \times \text{Glucose (mmol/L)}) \div 22.5$$

$$\text{Glucose : Insulin} = \text{Glucose (mg/dl)} \div \text{Insulin (uIU/mL)}$$

To convert glucose mmol/L to mg/dl, multiple by 18
To convert insulin pg/ml to uIU/mL, divide by 6

In periods one and three, six cats were successfully catheterized in each period, while in period two, only five cats were successfully catheterized. One cat was also excluded from plasma leptin concentration analyses due to an error in sample processing.

## Activity monitoring

Voluntary physical activity was measured over six consecutive days (d 15–21) during Study 1 using Actical® activity monitors (Mini Mitter, Bend, OR) that have been previously validated for use in cats [50]. Activity monitors were secured on the cats via harnesses that the cats had been acclimated. The period captured included Saturday and Sunday where less human interaction occurred as well as four days during the week where normal levels of human interaction occurred. Once the monitors were removed, Actical® software analyzed and converted the data into activity counts defined over a designated time period (15 s). In period one, three cats had defective activity monitors, and therefore no activity data was collected from those cats in that period. We considered daytime: nighttime activity to follow the 12 h light– 12 h dark cycle, with the lights turning on at 0800 h and off at 2000 h.

## Indirect calorimetry

Indirect calorimetry was performed on d 14 of each period in Study 2. These sessions began at approximately 0600h and were 24 h in length including: a 30-min respiratory gas equilibration, a fasted (1.5 h), fed and extended postprandial state (22 h). Indirect calorimetry was conducted by measuring respiratory gases for 5 min every 0.5 h. Concentrations of $O_2$ and $CO_2$ present in the respiratory chambers were measured with $O_2$ and $CO_2$ gas analyzers (Qubit Systems[®], Kingston, Ontario). The calorimeter used was an open circuit, ventilated calorimeter with room air being drawn through at a rate of ~5.5–6.5 L/min depending on cat weight and to ensure $CO_2$ did not exceed 0.7%. Airflow was set at 5.5–6.5 L/min, and the actual rate was measured with the use of a mass flow meter to measure total air volume in order to calculate and $VO_2$ and $VCO_2$. Gas analysers and mass flow meters were calibrated prior to each individual study and at least every 6 h during a study, or when a drift of more than 1% was observed. Calibration was performed using standard gas mixtures (nitrogen and carbon dioxide) at two concentrations (99.98% and 1.01%). RQ was calculated within the C950 Multi Channel Gas Exchange Software provided by Qubit Systems and EE was calculated as follows [51]:

$$EE \left(\frac{\text{kcal}}{\text{d}}\right) = \left[3.94 \times \text{O2 exchange}\left(\frac{\text{L}}{\text{h}}\right) + 1.11 \times \text{CO2 exchange}\left(\frac{\text{L}}{\text{h}}\right)\right] \times 24h$$

## Statistical analyses

All statistical analyses were performed using Statistical Analysis System (SAS, version 9.4; SAS Institute, Cary, NC, USA). Meal response data (appetite-regulating hormones, glucose, insulin, AA, EE and RQ) were analyzed using the MIXED procedure with repeated measures, where treatment (once a day feeding or four times a day feeding), sequence, and time (time within day around feeding) were fixed effects, time as a repeated measure, and cat within period as a random effect. Sequence had a significant effect on plasma concentrations of ghrelin and tyrosine, however, as sequence had no effect on the other appetite-regulating hormones or AA, sequence was not included in the final MODEL statement. Repeated measures were analysed using the covariance structure compound symmetry. Treatment least square means (LSM) were calculated and means were compared using the PDIFF multiple comparison procedure.

Area under the curve (AUC) for hormones, essential AA, as well as time periods (pre-prandial, 0–180 min, 180–540 min, 540–900 min, 900–1320 min) of RQ were calculated using GraphPad Prism version 8.01 for macOS (GraphPad Software). The MIXED procedure was used to analyze total activity, fed and fasted EE and RQ, fasted concentrations, and all AUC data using treatment and sequence as fixed effects and cat within period as the random effect. Total activity was the grouped by week day and weekend. Treatment least square means (LSM) were calculated using the LSMEANS statement and were compared using the PDIFF multiple comparison procedure treatment was significant. A $P < 0.05$ was considered significant and $0.05 > P < 0.10$ was considered a trend. All data are represented as LSM ± standard error of the mean (SEM).

# Results

## Body weight and feed intake

BW was similar among treatments and throughout both Study 1 and Study 2 (Table 2). In Study 1, there was no detected effect of treatment on energy intake (P = 0.685), while in Study 2, cats fed once daily ate less (P<0.001) compared to cats fed four times daily.

**Table 2. Body weight (kg) at baseline and weekly throughout Study 1 and Study 2 and energy intake (kcal/day) at baseline and weekly throughout Study 1 and Study 2 in cats eating a canned diet once (0800 h) or four (0800 h, 1130 h, 1500 h, 1830 h) times daily.**

| | | | One meal | Four meals | | | |
| | | | (n = 8)[1] | (n = 8)[1] | | | |
| | | Week | LSM[a] | LSM[a] | SEM[a] | $P_{treatment}$ | $P_{treatment \times week}$ |
|---|---|---|---|---|---|---|---|
| Study 1 | BW[a] (kg) | 0 | 4.31 | 4.36 | 0.03 | 0.063 | 0.868 |
| | | 1 | 4.30 | 4.34 | 0.03 | | |
| | | 2 | 4.30 | 4.33 | 0.03 | | |
| | | 3 | 4.29 | 4.31 | 0.03 | | |
| | Energy intake (kcal/day) | 0 | 190.05 | 193.16 | 5.31 | 0.685 | 0.680 |
| | | 1 | 192.66 | 196.68 | 5.31 | | |
| | | 2 | 188.73 | 190.22 | 5.31 | | |
| | | 3 | 191.80 | 187.69 | 5.31 | | |
| Study 2 | BW (kg) | 0 | 4.24 | 4.24 | 0.02 | 0.223 | 0.206 |
| | | 1 | 4.24 | 4.24 | 0.02 | | |
| | | 2 | 4.21 | 4.24 | 0.02 | | |
| | Energy intake (kcal/day) | 0 | 194.36 | 200.04 | 2.06 | < 0.001 | 0.961 |
| | | 1 | 195.34 | 200.30 | 1.88 | | |
| | | 2 | 195.84 | 199.76 | 2.00 | | |

[a] BW, body weight; LSM, least square means; SEM, standard error of the mean.

[1] Eight cats were used and 4 of 8 repeated one of the two treatments, each mean represents 12 experimental periods.

## Plasma appetite-regulating hormones

No significant differences between treatments were detected for fasted plasma concentrations for any of the appetite-regulating hormones; however, fasted ghrelin concentrations tended to be greater in cats fed once daily compared to cats fed four times per day (P = 0.058) (Table 3).

**Table 3. Fasted concentration and incremental area under the curve of appetite-regulating hormones in cats eating a canned diet once (0800 h) or four (0800 h, 1130 h, 1500 h, 1830 h) times daily.**

| | One meal | Four meals | SEM[a] | $P_{treatment}$ |
| | (n = 8)[1] | (n = 8)[1] | | |
| | LSM[a] | LSM[a] | | |
|---|---|---|---|---|
| **Fasted concentration** | | | | |
| Ghrelin (ng/ml) | 0.14 | 0.08 | 0.03 | 0.058 |
| GIP[a] (pmol/L) | 5.48 | 5.21 | 2.53 | 0.916 |
| GLP-1[a] (pmol/L) | 4.66 | 4.65 | 1.03 | 0.989 |
| Leptin (ng/ml) | 3.54 | 4.14 | 0.67 | 0.403 |
| PYY[a] (pmol/L) | 21.86 | 21.08 | 2.58 | 0.769 |
| **IAUC$_{0-24 h}$[a]** | | | | |
| Ghrelin (ng/ml x min) | 153.19 | 116.38 | 17.78 | 0.103 |
| GIP[a] (pmol/L x min) | 48042 | 43819 | 3685 | 0.282 |
| GLP-1[a] (pmol/L x min) | 13802 | 12348 | 2334 | 0.549 |
| Leptin (ng/ml x min) | 5580 | 5151 | 886.25 | 0.643 |
| PYY[a] (pmol/L x min) | 33426 | 32222 | 2422 | 0.631 |

[a] GIP, gastric inhibitory polypeptide; GLP-1, glucagon-like peptide-1; IAUC, incremental area under the curve; LSM, least square means; PYY, peptide-YY; SEM, standard error of the mean.

[1] Eight cats were used and 4 of 8 repeated one of the two treatments, each mean represents 11 experimental periods.

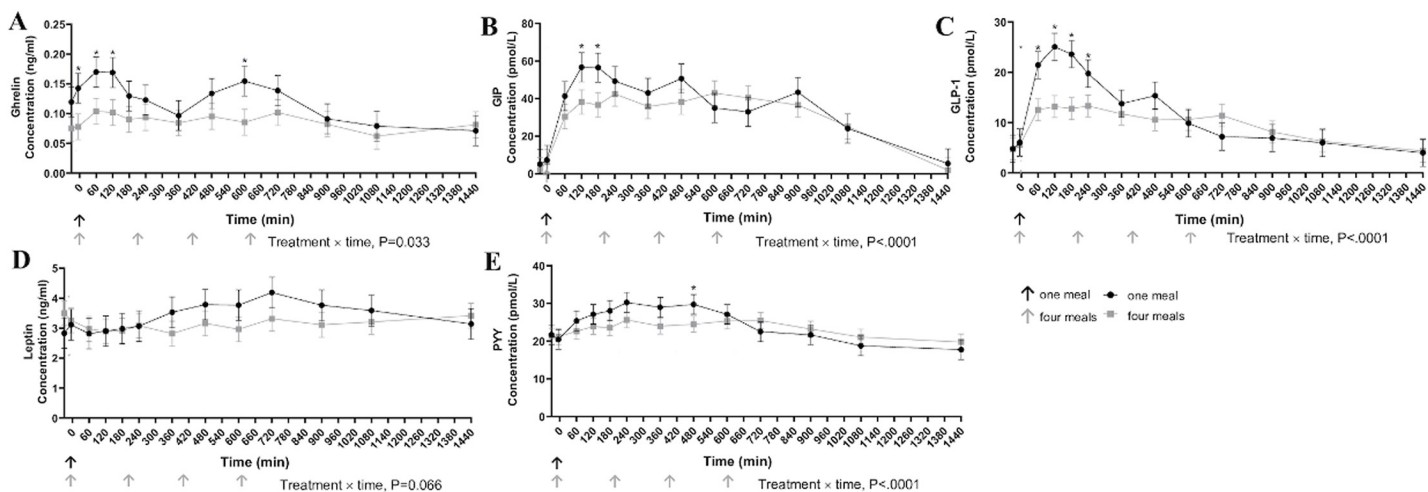

**Fig 1.** Plasma concentrations of ghrelin (ng/ml) (A), GIP (pmol/L) (B), GLP-1 (pmol/L) (C), leptin (ng/ml) (D), and PYY (pmol/L) (E) in cats eating a canned diet once (0800h) or four (0800h, 1130h, 1500h, and1830h) times daily over 24 hours. (Eight cats were used and 4 of 8 repeated one of the two treatments, each mean represents 11 experimental periods). Values are expressed as LSM ± SEM. * = P<0.05.

There was a significant treatment by time interaction (P< 0.05) and extended postprandial responses for plasma appetite-regulating hormones are shown in Fig 1. Note that timing starts after the provision of the only meal in cats fed once daily and the first meal of cats fed four times daily. Plasma ghrelin concentrations were greater at -5 min, and 60, 120, and 600 min post feeding in cats fed once per day compared to four times per day (P = 0.033). Cats fed once daily also had greater plasma GIP concentrations at 120 and 180 min post feeding (P<0.001) and greater concentrations of plasma GLP-1 at 60, 120, 180, and 240 min post feeding (P<0.001) in comparison to cats fed four times. No treatment by time effect was detected for plasma leptin concentrations, but there was a tendency for cats fed once daily to be greater than cats fed four times daily (P = 0.066). Plasma PYY concentrations were greater 480 min post feeding in cats fed once daily compared to cats fed four times daily (P<0.001).

No differences were detected for incremental AUC (IAUC) for any of the appetite-regulating hormones (Table 3).

## Whole blood glucose and plasma insulin

No differences between treatments were detected in fasted whole blood glucose and plasma insulin concentrations (Table 4).

Whole blood glucose response was greater 24 h (1440 min) post feeding in cats fed once per day (P < .0001) in contrast to cats fed four times (Fig 2). Cats fed once daily had greater plasma concentrations of insulin at 60 and 120 min post feeding (P < .001). The glucose to insulin ratio was lower 5 min prior to the first meal in cats fed once a day (P < .001). Cats fed once daily had a greater HOMA-IR 60 and 120 min after the meal (P<0.001). Incremental AUC for glucose and insulin did not differ between treatments (Table 4).

## Plasma amino acids

Fasted concentrations of isoleucine (P = 0.038) were lesser and concentrations of leucine (P = 0.097) and valine (P = 0.06) tended to be lesser in cats fed once daily compared to four times per day. No other differences were detected in fasted concentrations of any other essential AA (Table 5).

**Table 4. Fasted concentration and incremental area under the curve of glucose and insulin in cats eating a canned diet once (0800 h) or four (0800 h, 1130 h, 1500 h, 1830 h) times daily.**

| | One meal | Four meals | SEM[a] | P$_{treatment}$ |
|---|---|---|---|---|
| | (n = 8)[1] | (n = 8)[1] | | |
| | LSM[a] | LSM[a] | | |
| **Fasted concentration** | | | | |
| Glucose (mmol/L) | 4.73 | 4.77 | 0.13 | 0.818 |
| Insulin (pmol/L) | 41.28 | 38.40 | 4.67 | 0.553 |
| **IAUC$_{0-24\ h}$[a]** | | | | |
| Glucose (mmol/L x min) | 7076 | 6936 | 102.88 | 0.216 |
| Insulin (pmol/L x min) | 77666 | 71743 | 4325 | 0.204 |

[a]IAUC, incremental area under the curve; LSM, least square means; SEM, standard error of the mean.

[1] Eight cats were used and 4 of 8 repeated one of the two treatments, each mean represents 11 experimental periods.

Meal responses for individual plasma AA are shown in Figs 3–5. Plasma glycine concentrations were greater in cats fed four times per day 360, 480, and 720 min after the first meal (P<0.001). Cats fed once daily had greater concentrations of alanine, arginine, asparagine, lysine, methionine, and serine 60 min following the meal (P<0.05). Alanine (P<0.001) and asparagine (P<0.001) concentrations were greater in cats fed once daily at 600, 720, and 900 min after the meal compared to cats fed four times per day. Average total essential amino acid concentrations were greater in cats fed once a day at 60 and 240 min post feeding and in contrast to cats fed four times a day (Fig 5). Despite differences in individual times points, IAUC did not differ for any of the essential AA (Table 5).

## Voluntary physical activity

Voluntary physical activity is reported in Table 6. Cats fed four times daily had greater total activity counts (P = 0.048) and average daily activity (P = 0.043) compared to cats fed once

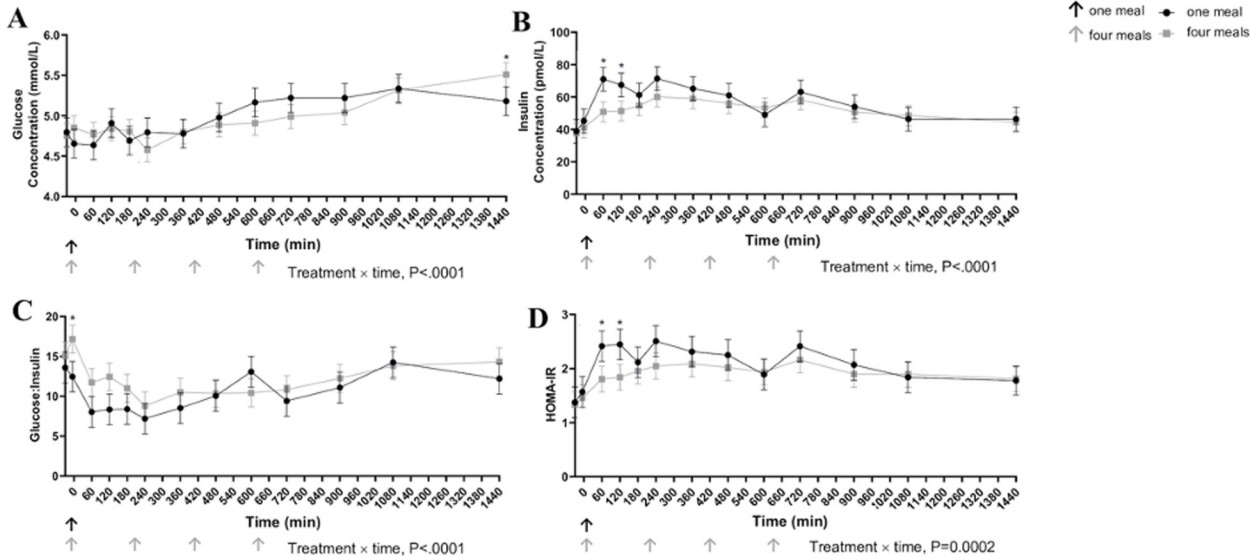

**Fig 2.** Whole glucose concentrations (mmol/L) (A), plasma insulin concentrations (pmol/L) (B), glucose to insulin ratio (C), and homeostatic model assessment of insulin resistance (HOMA-IR) (D) in cats eating a canned diet once (0800h) or four (0800h, 1130h, 1500h, and1830h) times daily over 24 hours (Eight cats were used and 4 of 8 repeated one of the two treatments, each mean represents 11 experimental periods). Values are expressed as LSM ± SEM. * = P<0.05.

**Table 5. Fasted concentration and incremental area under the curve of amino acids in cats eating once (0800 h) or four (0800 h, 1130 h, 1500 h, 1830 h) times daily.**

| | One meal (n = 8)[1] LSM[a] | Four meals (n = 8)[1] LSM[a] | SEM[a] | P_treatment |
|---|---|---|---|---|
| **Fasted concentration (uM)** | | | | |
| Arginine | 147.11 | 137.98 | 11.21 | 0.436 |
| Histidine | 103.84 | 114.21 | 7.59 | 0.205 |
| Isoleucine | 82.01 | 92.26 | 4.20 | 0.038 |
| Leucine | 148.81 | 162.29 | 7.29 | 0.097 |
| Lysine | 157.94 | 142.59 | 10.41 | 0.174 |
| Methionine | 37.86 | 39.90 | 4.24 | 0.641 |
| Phenylalanine | 60.79 | 65.87 | 3.68 | 0.200 |
| Taurine | 67.00 | 49.80 | 11.75 | 0.177 |
| Threonine | 127.43 | 114.90 | 13.10 | 0.364 |
| Tryptophan | 56.56 | 56.67 | 5.92 | 0.985 |
| Valine | 211.20 | 235.55 | 11.30 | 0.06 |
| **IAUC$_{0-24 h}$[a] (uM x min)** | | | | |
| Arginine | 4172 | 3913 | 217.30 | 0.263 |
| Histidine | 2884 | 2951 | 134.42 | 0.631 |
| Isoleucine | 2109 | 2089 | 71.63 | 0.785 |
| Leucine | 4330 | 4244 | 187.17 | 0.656 |
| Lysine | 4024 | 4147 | 213.58 | 0.580 |
| Methionine | 1925 | 2023 | 146.17 | 0.518 |
| Phenylalanine | 1405 | 1417 | 82.86 | 0.893 |
| Taurine | 1892 | 1992 | 164.74 | 0.559 |
| Threonine | 3864 | 3570 | 275.39 | 0.314 |
| Tryptophan | 1381 | 1451 | 97.78 | 0.490 |
| Valine | 6139 | 6066 | 233.53 | 0.763 |

[a]LSM, least square means; SEM, standard error of the mean; IAUC, incremental area under the curve.

[1] Eight cats were used and 4 of 8 repeated one of the two treatments, each mean represents 11 experimental periods.

daily. Average activity during the daylight hours (P = 0.007) and light: dark activity ratios (P<0.001) were also greater in cats fed four times, while cats fed once daily were more active during the dark hours (P = 0.035). Cats eating four meals per day were more active during weekdays (P = 0.022) compared to cats consuming one meal daily. No differences were detected in activity on the weekend (P = 0.987).

## Indirect calorimetry

Fasted EE was similar between cats consuming one or four meals daily (P = 0.470). Similarly, feeding regimen had no detected effect on average postprandial EE (P = 0.612). Fasted RQ was lower (P = 0.004) in cats fed once daily compared to cats fed four times daily.

Treatment had no detected effect on average postprandial RQ (P = 0.912) (Table 7). Treatment × time affected EE over the entire 24-hr period (P = 0.027) (Fig 6). A treatment × time effect was observed where cats fed once per day had a lower RQ 90–180 min and 1230–1320 min after the meal compared to cats fed four times daily (P<0.001). Cats fed four times daily had a lesser RQ 720 and 820–890 min after the first meal (P<0.001). No differences were detected on AUC for RQ between feeding regimens (P>0.05) (Table 7).

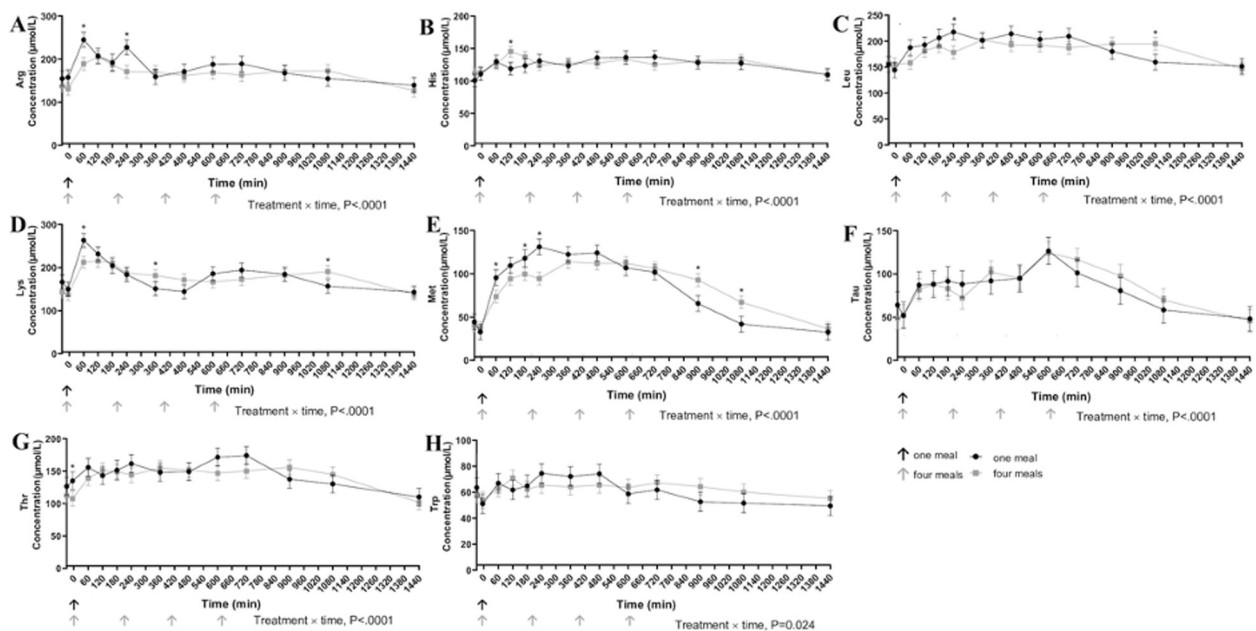

**Fig 3.** Plasma concentration (μmol/L) of arginine (A), histidine (B), leucine (C), lysine (D), methionine (E), taurine (F), threonine (G), and tryptophan (H) in cats eating a canned diet once (0800h) or four (0800h, 1130h, 1500h, 1830h) times daily over 24 hours (n = 8 but with 11 experimental periods). Values are expressed as LSM ± SEM. * = P<0.05.

## Discussion

Meal frequency has been studied in humans as well as animal models, and intermittent fasting may result in increased protein synthesis, lower RQ, and reduced fat mass [23, 29]. Some

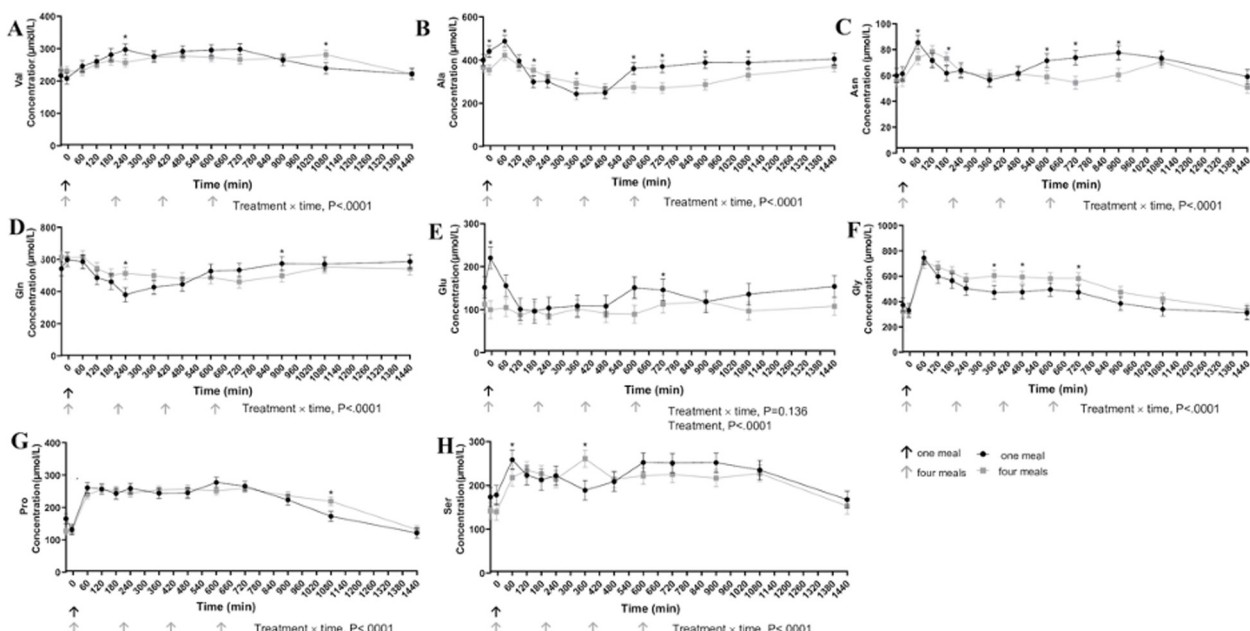

**Fig 4.** Plasma concentration (μmol/L) of valine (A), alanine (B), asparagine (C), glutamine (D), glutamate (E), glycine (F), proline (G) and serine (H) in cats eating a canned diet once (0800h) or four (0800h, 1130h, 1500h, and 1830h) times daily over 24 hours (n = 8 but with 11 experimental periods). Values are expressed as LSM ± SEM. * = P<0.05.

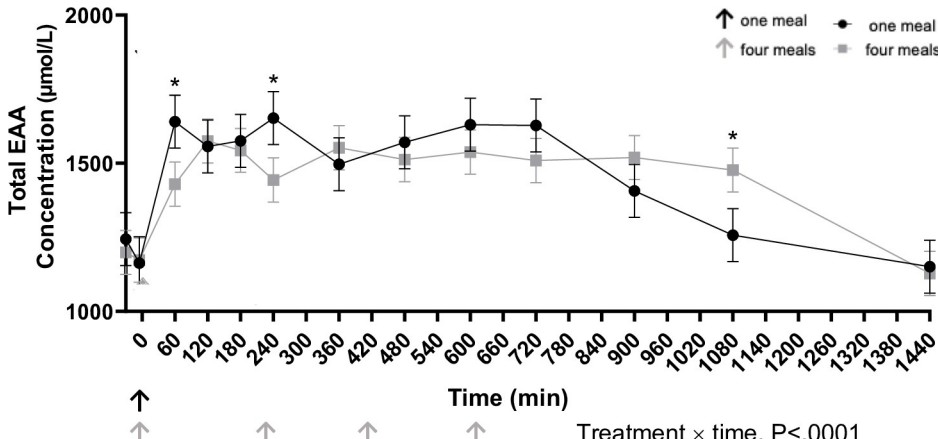

**Fig 5. Total plasma essential amino acid concentration (μmol/L) in cats eating a canned diet once (0800h) or four (0800h, 1130h, 1500h, and 1830h) times daily over 24 hours (n = 8 but with 11 experimental periods).** Values are expressed as LSM ± SEM. * = P<0.05

researchers have suggested that consuming small meals more frequently is preferred to control appetite and increase physical activity, especially in cats [19, 20]. However, there is a paucity of literature that defines which feeding regimen is most beneficial for cats with regards to satiation, LBM maintenance, and fat mass reduction. The present study was the first to evaluate how feeding regimens affect cats' appetite-regulating hormones (GIP, GLP-1 and PYY), amino acid metabolism, EE, and RQ over a 24-hour sampling period. The results of this study suggest that cats fed once per day were more satiated than those fed four times a day, as they had greater postprandial levels of appetite-regulating hormones, GIP, GLP-1, and PYY. Cats fed once daily had higher levels of the orexigenic hormone, ghrelin, that may have been due to an anticipatory response prior to meal times [52–54]. Lower RQ following the meal in cats fed once daily suggests greater fat oxidation instead of carbohydrate oxidation [55], similar to humans following an intermittent fasting routine who had lower RQs and, therefore, lesser fat mass [23]. Greater postprandial increases in circulating concentrations of several AA in cats fed once daily may suggest a greater potential for protein synthesis, similar to what has been reported in neonatal animals and resistance-training adult humans [29, 32, 33].

**Table 6. Voluntary physical activity (arbitrary activity counts) of cats eating once (0800 h) or four times (0800 h, 1130 h, 1500 h, 1830 h) daily consisting of six consecutive days of data for each treatment.**

|  | One meal | Four meals | SEM[a] | P[treatment] |
|---|---|---|---|---|
|  | (n = 8)[1] | (n = 8)[1] |  |  |
|  | LSM[a] | LSM[a] |  |  |
| Total arbitrary activity counts | 240651 | 281337 | 18491 | 0.048 |
| Average Daily Activity | 7.21 | 8.43 | 0.56 | 0.043 |
| Light Average | 8.61 | 12.11 | 0.97 | 0.007 |
| Dark Average | 6.19 | 5.36 | 0.30 | 0.035 |
| Average Daily Light: Dark Activity | 1.51 | 2.86 | 0.32 | <0.001 |
| Weekdays | 7.52 | 9.61 | 0.03 | 0.022 |
| Weekend | 6.44 | 6.43 | 0.85 | 0.987 |

[a]LSM, least square means; SEM, standard error of the mean.

[1] Eight cats were used and 4 of 8 repeated one of the two treatments, each mean represents 12 experimental periods.

**Table 7. Energy expenditure (kcal/kg BW), respiratory quotient, and area under the curve for the respiratory quotient of cats eating once (0800 h) or four times (0800 h, 1130 h, 1500 h, 1830 h) daily over a 24-h period.**

| | One meal | Four meals | SEM[a] | P_treatment |
|---|---|---|---|---|
| | (n = 8)[1] | (n = 8)[1] | | |
| | LSM[a] | LSM[a] | | |
| EE[a] fasted (kcal/kg BW) | 32.72 | 36.11 | 4.57 | 0.470 |
| EE[a] fed (kcal/kg BW) | 38.39 | 36.27 | 4.10 | 0.612 |
| RQ[a] fasted | 0.68 | 0.70 | 0.01 | 0.004 |
| RQ[a] fed | 0.68 | 0.68 | 0.01 | 0.912 |
| **Time around feeding AUC[a] (RQ[a] x min)** | | | | |
| Pre-prandial | 61.44 | 58.98 | 3.36 | 0.474 |
| 0–180 min | 117.92 | 119.92 | 1.82 | 0.288 |
| 180–540 min | 245.27 | 244.06 | 2.5 | 0.635 |
| 540–900 min | 251.47 | 246.53 | 3.75 | 0.209 |
| 900–1320 min | 284.94 | 291.56 | 3.81 | 0.103 |

[a]AUC, area under the curve; BW, body weight; EE, energy expenditure; LSM, least square means; RQ, respiratory quotient; SEM, standard error of the mean.

[1] Eight cats were used and 4 of 8 repeated one of the two treatments, each mean represents 11 experimental periods.

Many circulating appetite-regulating hormones are anorexigenic and are associated with satiety [56]. Lower circulating ghrelin in cats fed four times daily agrees with previous research where plasma ghrelin IAUC remained lower in cats fed four times daily compared to cats fed two times daily [19]. Similar low concentrations of ghrelin were also noted in humans with greater meal frequency [56] and men consuming four meals each hour throughout the morning, compared to consuming one meal in the morning [57]. Only Leidy et al. [58] did not detect any differences in ghrelin concentrations in humans consuming three versus six meals over an 11-hour period. However, as the remaining satiety hormones in the present study suggest that cats fed once daily were more satiated; increased levels of ghrelin may be due to meal anticipation, as seen previously in rats and humans [52–54] and deserves further investigation to understand the relationship between ghrelin and anticipatory feeding behaviour in cats.

Leptin is an anorexigenic hormone that contributes to the chronic regulation of food intake [59], causing the feeling of hunger to subside [60]. In a previous study, plasma leptin IAUC were higher in cats consuming four meals daily compared to cats fed two times per day [19]. Higher serum leptin concentrations were also observed in horses fed four times daily, compared to three or two times daily [61]. In contrast, leptin concentrations did not differ between type 2 diabetes patients consuming either two meals or six meals daily [62] and, similarly, leptin was not affected by treatment in the present study. Further investigation is warranted to resolve the seemingly opposite results of Deng et al. [19] and the current study on how single, double, or multiple meals may differentially affect the satiety hormone response.

In humans, GLP-1 is known as an incretin hormone and mediates about 80% of insulin secreted, alongside GIP, after glucose is ingested [63]. Glucagon-like peptide-1 response in cats is similar to humans, where basal GLP-1 serum concentrations are similar with comparable postprandial increases [64]. While a postprandial increase in GLP-1 has been observed in cats in response to a meal [64–66]; this is the first study to demonstrate that cats fed once per day had greater concentrations of GLP-1 in contrast to four times per day. Similarly, consuming a large caloric load caused a greater release in GLP-1 and GIP in humans compared to consuming several small meals over time [67, 68]. Lindgren et al. [69] also reported a more pronounced release of GLP-1 after the morning meal compared to the meal consumed in the afternoon in lean healthy adult males. These results are similar to the present study, where a

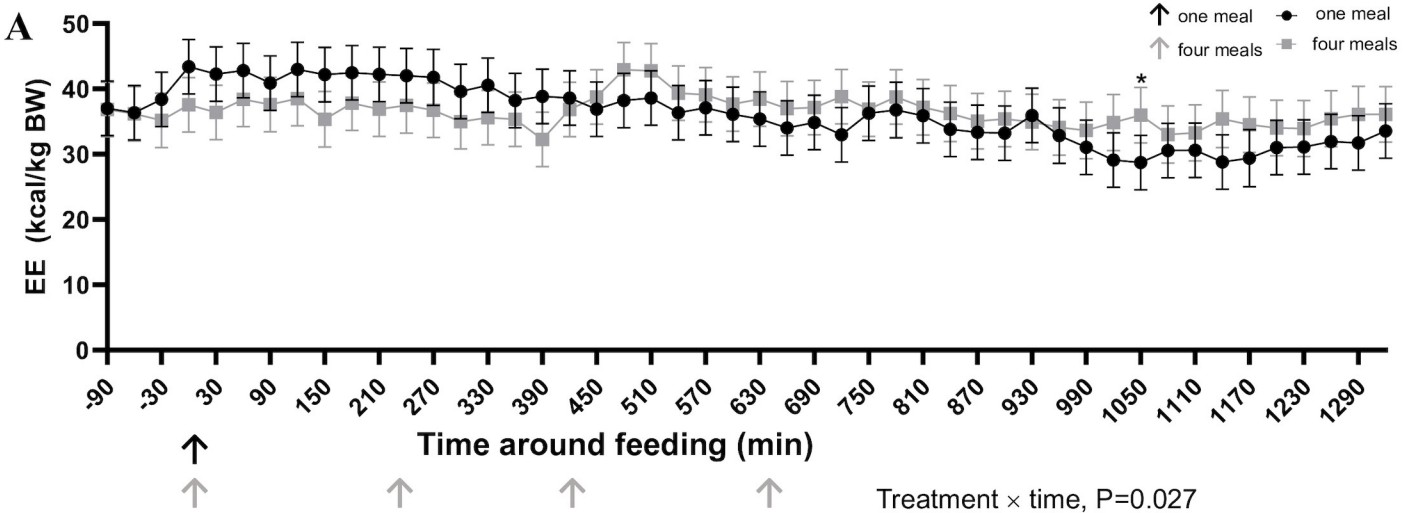

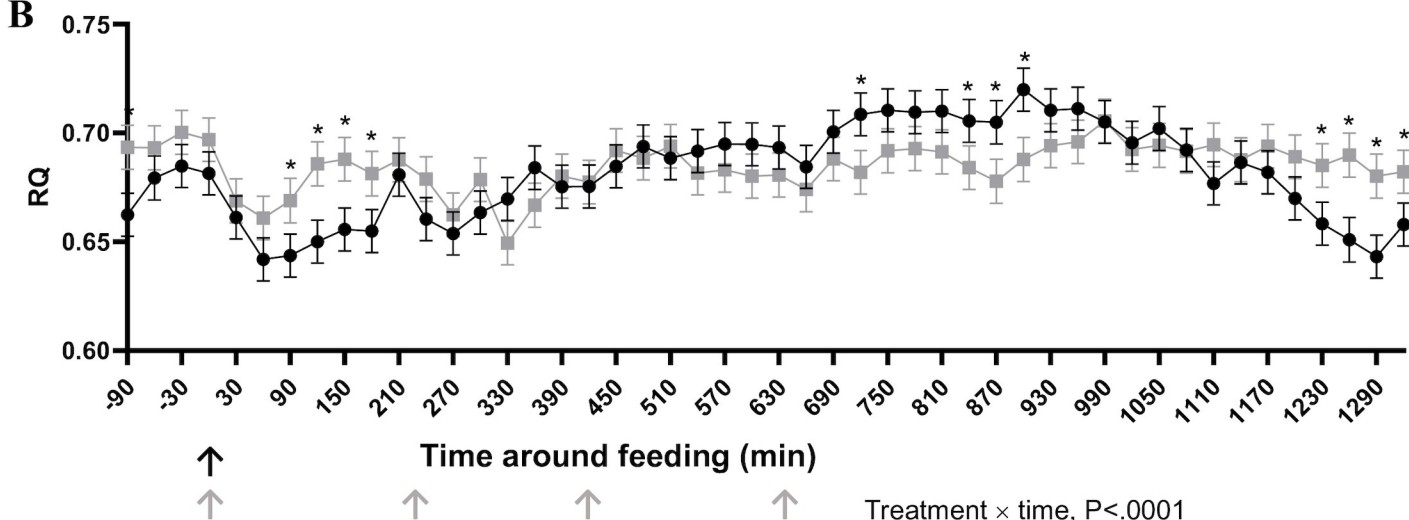

**Fig 6.** Energy expenditure (EE) (kcal/kg BW) (A) and respiratory quotient (RQ) (B) in cats eating a canned diet once (0800h) or four (0800h, 1130h, 1500h, and 1830h) times daily over 24 hours (n = 8 but with 12 experimental periods). Values are expressed as LSM ± SEM. * = P<0.05.

greater postprandial GLP-1 increase occurred in cats fed once daily in the morning compared to cats fed four times daily, receiving two of their meals in the afternoon. This could be due to the greater caloric intake that cats fed once daily received per meal or as part of a circadian rhythm and, thus, warrants further investigation. The greater increase in GLP-1 in response to the larger caloric load cats fed once daily consumed may suggest larger meals are needed to stimulate the release of large amounts of this hormone into circulation [70]. Overall, the present study demonstrates similar results to studies performed in humans [67, 68] where a larger caloric load resulted in a greater postprandial GLP-1 response.

The role of GIP in cats may differ from that of other species, as cats our obligate carnivores and possess numerous metabolic idiosyncrasies. Gilor et al. [64] reported that GIP concentrations did not increase in cats after an oral glucose tolerance test, but did increase after being stimulated with a bolus of lipids or AA. Therefore, the role of GIP in cats may not equate to

the role as an incretin hormone in humans, but still appears to play a role in mediating satiety. Similar to previously published studies performed with cats [66, 71], the present study demonstrated that GIP concentrations increase postprandially. In humans, plasma GIP concentrations remained near fasting over the duration of a three day study and showed no variability in subjects consuming six meals daily [72]. However, in the same study, subjects consuming three meals daily had greater meal responses in plasma GIP concentrations [72]. In the present study, cats consuming four meals daily experienced an initial postprandial increase in GIP concentrations and then showed no change throughout the day. Lindgren et al. [69] reported a greater GIP response after the morning meal compared to the afternoon meal in lean healthy males. This is similar to the present study, as GIP concentrations peaked after the first meal in cats fed once and four times daily. However, no further increase was observed in cats fed four times daily after their additional meals (1130 h, 1500 h, 1830 h), compared to the initial peak of GIP after the first meal (0800 h). Therefore, it appears that GIP patterns in cats may follow a similar pattern to humans, as greater increases were observed in the morning after the first meal, compared to meals consumed in the afternoon.

Peptide YY is a hormone released in response to meals, and its reduction encourages hunger cascades in humans [73]. The physiology of PYY is still unknown in cats, but is known to increase due to meal feeding similar to other species [66]. In agreement with the current findings, Degen et al. [74] observed a larger and sustained increase in PYY levels in humans after a large meal was ingested compared to a smaller meal of the same composition, but different caloric content. Concentrations of PYY were lesser in response to increased compared to reduced eating frequency (six versus three meals) in humans [75], which agrees with the present study. The greater increase in PYY in response to the cats being fed one larger meal per day may suggest that larger meals preferentially stimulate the release of large amounts of PYY into circulation [74]. Similar to previous research done with humans, larger meals appear to cause a greater response in plasma PYY concentrations in cats, likely reducing hunger.

In the present study, whole blood glucose did not differ between cats eating once or four times daily throughout the 24-hour period. In previous studies, glucose levels did respond in cats fed four compared to two meals daily [19], and in humans receiving six versus three meals per day [55, 75]. However, other researchers reported that blood glucose levels did not differ between cats fed the same dry extruded diet *ad libitum* or once daily [76]. Gluconeogenesis is the synthesis of glucose from lactate, pyruvate, glycerol, and the carbon backbone of several gluconeogenic amino acids. In cats, activity of enzymes involved in gluconeogenesis is greater than in other mammals [77, 78] and gluconeogenesis always occurs in the liver [79, 80]. Interestingly, and particularly in cats fed once per day, beginning 600 min after the meal, glucose concentrations follow a parallel pattern to glucogenic amino acids; arginine, alanine, asparagine, glutamine, glutamate and serine. This is further supported by the increase in RQ at this time, suggesting a shift towards glucose oxidation [47]. Similar to Asaro et al. [81], where interstitial glucose levels followed a pattern similar to that of RQ, whole blood glucose concentrations in the present study appeared to parallel RQ as well, and may be further related to the availability of glucogenic amino acids for gluconeogenesis.

In the present study cats fed once daily had a greater increase in plasma insulin concentrations following a meal than cats fed four times per day. Researchers have previously detected more static concentrations of plasma insulin in cats fed four meals daily compared to cats fed two meals daily [19] and in humans consuming more frequent meals [57]. These findings are similar to the insulin concentrations noted in cats fed four meals in the present study. Amino acids, particularly arginine, have been demonstrated to stimulate insulin secretion in cats [82]. Further, greater increases in plasma insulin concentrations in cats fed once daily may stimulate muscle protein synthesis similar to neonatal animals [83], as insulin stimulates energy sensing

and protein synthetic pathways. A greater increase in plasma AA was also observed in cats fed one meal daily, corresponding to this peak in plasma insulin concentrations. It appears that this peak in AA, rather than glucose, stimulated insulin secretion, similar to that observed in a previous study performed with cats [84]. This is also further substantiated by the lower RQ seen in cats fed once daily following their meal, indicating that the cats were oxidizing proportionately more fat [47].

Plasma AA concentrations were generally similar to pre-prandial and postprandial concentrations reported previously in cats [85–87]; however, the present study is the first to report plasma AA concentrations over a 24 hour period in cats. Several AA (alanine, arginine, asparagine, glutamate, leucine, lysine, methionine, proline, serine, threonine, and valine) had greater increases in cats fed once daily and could be important for muscle protein synthesis and maintenance of lean tissue mass as demonstrated in pigs [29, 32]. In neonatal piglets, an increase in branched chain amino acids 60 min after an intermittent bolus meal activated the signaling components that regulate translation initiation and stimulate protein synthesis [29]. Ninety minutes after a bolus meal, muscle protein synthesis in intermittently fed neonatal pigs was twice that of continuously fed piglets [32]. These results support the hypothesis that intermittent fasting enhances lean tissue accumulation to a greater degree than continuous feedings in neonatal piglets. Similarly, in the present study, for those cats fed once daily a greater increase in essential AA 60 min after the meal was observed in comparison to cats fed four times daily. In healthy young men, the consumption of an oral bolus of essential AA results in increased muscle protein synthesis 45–90 min after consuming an oral bolus of essential AA [88, 89]. When AA are minimally increased and remain constant, protein synthesis is only modestly stimulated, therefore, it appears the bolus of AA within a single meal is required to maximally stimulate protein synthesis [32]. Together, it is possible that this bolus in AA, as well as insulin, in cats fed once daily may be needed to maximally stimulate protein synthesis. Aging cats may also benefit from a feeding regimen that resembles intermittent fasting, as consuming large protein dense meals has resulted in more AA being absorbed and used for muscle protein synthesis in older men [90]. However, whole body protein synthesis, tissue protein syntheis, or protein synthesis biomarkers were not evaluated in the present study to determine if there was a peak in translation and protein synthesis. Future work could employ a variety of methods to examine the effects of feeding frequency on protein synthesis in cats.

Similar to past research, total activity count and average daily activity were greater in cats fed four times daily [20, 21]. Similar to the studies by de Godoy et al. [20] and Deng et al. [21], greater activity was recorded during the daylight hours in cats eating four meals, and the ratio of light: dark activity was also greater compared to cats eating once daily. Overall, the present study confirmed that multiple meal feeding increases voluntary physical activity during the daylight hours. Building on previous work and hypothesizing that cats fed once per day were more satiated, increased physical activity during the day for cats fed fours times per day could be a result of hunger motivating the cats to be more active and engaged their environment in an attempt to be fed [91]; this, in turn, may have led them to consume more calories as suggested by the results from Study 2.

The present study is the first to directly assess the effects of feeding regimen on EE and RQ in cats. Previous studies investigating the RQ of cats by the use of indirect calorimetry have fed a dry extruded diet [81, 92–94]. Therefore, the RQs presented herein are lower than previous literature, as the diet fed in the present study was a canned cat food with relatively low carbohydrate concentrations in contrast to previous studies. Cats fed once daily had a lower fasted RQ compared to cats fed four times daily. Individuals with a lower RQ are theoretically protected against future fat accumulation [95] and RQ below 0.70 suggest that gluconeogenesis is occurring. In one study investigating the effects of intermittent fasting during Ramadan (strict

fasting is observed from sunrise to sunset for four weeks) it was discovered that resting metabolic rate was lower after the first week of IF during Ramadan [96]. The same study reported a lower RQ during Ramadan as compared to after. These results are similar to other studies by el Ati et al. [97] and Antoni et al. [98], suggesting a shift towards fat metabolism rather than carbohydrate as a source of metabolic energy during IF. Moro et al. [23] reported a decrease in fat mass and RQ in healthy resistance-trained males after eight weeks of practicing IF, whereas resting EE did not change between the IF group and the normal diet group. In fact, several studies investigating IF in humans detected no changes in EE [99–101]. Therefore, taken together, cats fed four times per day may have greater EE, as they were more active during the day, but this is seemingly offset by the nutrient partitioning in cats fed once per day.

## Conclusions

Overall, feeding cats once per day presents several promising outcomes to improve the quality of life of indoor cats, as feeding regimen could reduce the incidence of obesity in cats, by controlling appetite and limiting feed intake. Such a feeding regimen could also improve protein synthesis, by increasing plasma AA and insulin and may be useful to combat sarcopenia in aging cats by increasing LBM. Consuming one meal per day caused cats to have a greater and more sustained response in appetite-regulating hormones GLP-1, GIP, and PYY, suggesting that these cats were more satiated than cats consuming smaller and more frequent meals. Cats fed once per day also consumed less food and had lower fasting RQs than cats eating meals more frequently, suggesting that over time, this feeding regimen could support weight loss and lower fat mass. A greater increase in plasma AA concentrations in cats fed once per day could also be indicative of greater protein synthesis and, in turn, could support LBM maintenance.

## Acknowledgments

We would like to thank all student volunteers for their help with animal handling and blood collections, as well as cleaning and feeding.

## Author Contributions

**Conceptualization:** Alexandra Camara, Adronie Verbrugghe, Trevor J. DeVries, Anna K. Shoveller.

**Data curation:** Alexandra Camara, Adronie Verbrugghe, Cara Cargo-Froom, Anna K. Shoveller.

**Formal analysis:** Alexandra Camara, Adronie Verbrugghe, Cara Cargo-Froom, Trevor J. DeVries, Lindsay E. Robinson, Anna K. Shoveller.

**Funding acquisition:** Alexandra Camara, Adronie Verbrugghe, Anna K. Shoveller.

**Investigation:** Alexandra Camara, Kylie Hogan, Andrea Sanchez, Lindsay E. Robinson, Anna K. Shoveller.

**Methodology:** Alexandra Camara, Adronie Verbrugghe, Kylie Hogan, Trevor J. DeVries, Andrea Sanchez, Lindsay E. Robinson, Anna K. Shoveller.

**Project administration:** Alexandra Camara, Lindsay E. Robinson, Anna K. Shoveller.

**Resources:** Alexandra Camara, Adronie Verbrugghe, Andrea Sanchez, Lindsay E. Robinson, Anna K. Shoveller.

**Software:** Alexandra Camara, Adronie Verbrugghe, Cara Cargo-Froom, Anna K. Shoveller.

**Supervision:** Adronie Verbrugghe, Anna K. Shoveller.

**Validation:** Alexandra Camara, Adronie Verbrugghe, Trevor J. DeVries, Lindsay E. Robinson, Anna K. Shoveller.

**Visualization:** Alexandra Camara, Adronie Verbrugghe, Cara Cargo-Froom, Trevor J. DeVries, Lindsay E. Robinson, Anna K. Shoveller.

**Writing – original draft:** Alexandra Camara, Adronie Verbrugghe, Kylie Hogan, Trevor J. DeVries, Lindsay E. Robinson, Anna K. Shoveller.

**Writing – review & editing:** Alexandra Camara, Adronie Verbrugghe, Cara Cargo-Froom, Kylie Hogan, Trevor J. DeVries, Andrea Sanchez, Lindsay E. Robinson, Anna K. Shoveller.

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
