## [Decision Letter · Decision Letter 0]

17 Jul 2020

PONE-D-20-15734

The daytime feeding frequency affects appetite-regulating hormones, amino acids, physical activity, and respiratory quotient, but not energy expenditure, in adult cats fed regimens for 21 days

PLOS ONE

Dear Dr. Shoveller,

Thank you for submitting your manuscript to PLOS ONE. After careful consideration, we feel that it has merit but does not fully meet PLOS ONE’s publication criteria as it currently stands. Therefore, we invite you to submit a revised version of the manuscript that addresses the points raised during the review process.

We look forward to receiving your revised manuscript.

Kind regards,

Juan J Loor

Academic Editor

PLOS ONE

Journal Requirements:

2.Thank you for stating the following in the Competing Interests section:

[The authors declare no conflicts of interest. A.V. is the Royal Canin Veterinary Diets Endowed Chair in Canine and Feline Clinical Nutrition at the Ontario Veterinary College.].

Reviewers' comments:

Reviewer's Responses to Questions

**Comments to the Author**

1. Is the manuscript technically sound, and do the data support the conclusions?

Reviewer #1: Yes

2. Has the statistical analysis been performed appropriately and rigorously? 

Reviewer #1: Yes

3. Have the authors made all data underlying the findings in their manuscript fully available?

Reviewer #1: Yes

4. Is the manuscript presented in an intelligible fashion and written in standard English?

Reviewer #1: Yes

5. Review Comments to the Author

Reviewer #1: PONE-D-20-15734 The daytime feeding frequency affects appetite-regulating hormones, amino acids, physical activity, and respiratory quotient, but not energy expenditure, in adult cats fed regimens for 21 days

I found the paper to be ab excellent contribution to the area. It was thorough and encompassing. My concerns were a lack of understanding of the experimental design and an apparent liberal application of statistics. However, the data are excellent despite a rather limited number of experimental observations.

Line

57 Delete been

60 Delete should

134 Change “recorded in grams (orts)”. To ‘orts recorded’

150 ‘2 x 3 replicated incomplete Latin square design’ I can say with complete confidence that this is one I have never heard of. I have read this multiple times and just cannot grasp all that was done. You describe an experiment that has periods and then you describe study 1 and study 2. What is a study? What in this relates to 2x3? I suspect it is not as Latin square bur is a crossover design.

188 You should name the actual additives.

190 Delete and to

278 I did not understand much about the design, I know you cannot have 8 cats and an n=12. Just not possible.

293 You should probably point out that there were Time x Trt effects that make these comparisons relevant as opposed to Trt effects which are not.

535 I do not see anything that gives relevance to this carbohydrate ceiling discussion. I would delete it.

560 A lot of discussion for RQ of 0.68 vs 0.70. Is this really significant or perhaps driven by the intake differences? Point is the discussion could be shortened some and this might be an opportunity.

6. PLOS authors have the option to publish the peer review history of their article (what does this mean?). If published, this will include your full peer review and any attached files.

Reviewer #1: No

---

## [Author Response · Author response to Decision Letter 0]

13 Aug 2020

Reviewer #1: PONE-D-20-15734 The daytime feeding frequency affects appetite-regulating hormones, amino acids, physical activity, and respiratory quotient, but not energy expenditure, in adult cats fed regimens for 21 days

I found the paper to be ab excellent contribution to the area. It was thorough and encompassing. My concerns were a lack of understanding of the experimental design and an apparent liberal application of statistics. However, the data are excellent despite a rather limited number of experimental observations.

Response: Thank you for your kind words. We too are excited about this data. Thank you for your patience while we found time to revise our manuscript.

Line

57 Delete been

Response: Completed.

60 Delete should

Response: Completed.

134 Change “recorded in grams (orts)”. To ‘orts recorded’

Response: Completed.

150 ‘2 x 3 replicated incomplete Latin square design’ I can say with complete confidence that this is one I have never heard of. I have read this multiple times and just cannot grasp all that was done. You describe an experiment that has periods and then you describe study 1 and study 2. What is a study? What in this relates to 2x3? I suspect it is not as Latin square bur is a crossover design.

Response: Latin square designs are commonly used in animal science and agricultural research and each treatment is equally represented in each period. Latin square designs are similar to cross-over designs, but ensure that treatments are equally allocated within period and blocking criteria (sex) is equally represented within treatments. We have added the following explanation in the text in the Experimental Design section:

“Meal frequency was tested in two separate studies, each using a 2 x 3 replicated incomplete Latin square design with two treatments (one vs. four times feeding frequency) and three periods. The third period allowed us to account for carry over effects of feeding frequency in such that we sought to understand whether a longer time receiving a feeding frequency altered our physiological response criteria. Because all cats repeated one of two treatments, this resulted in 12 experimental periods per treatment as four out of eight cats repeated one of two treatments.”

188 You should name the actual additives.

Response: We have added the full name, Dipeptidyl peptidase-IV inhibitor.

190 Delete and to

Response: Deleted.

278 I did not understand much about the design, I know you cannot have 8 cats and an n=12. Just not possible.

Response: We included the repeated period that 50% of the cats did on each treatment, resulting in an n=12. To make this more clear we have added Number of subjects (n=8) and experimental periods (11 or 12 depending on outcome) to the tables and figures.

293 You should probably point out that there were Time x Trt effects that make these comparisons relevant as opposed to Trt effects which are not.

Response: Great point, thank you.

535 I do not see anything that gives relevance to this carbohydrate ceiling discussion. I would delete it.

Response: We had included this as the rationale for the lower voluntary food intake noted in cats fed once a day, but agree that it likely does not add much to the discussion and have deleted that paragraph.

560 A lot of discussion for RQ of 0.68 vs 0.70. Is this really significant or perhaps driven by the intake differences? Point is the discussion could be shortened some and this might be an opportunity.

Response: Indirect calorimetry is very precise and changes in macronutrient partitioning underpin respiratory quotients. Because the current results suggest an increase in lipid utilization for energy production and are clinically relevant to weight loss strategies, as such, we would prefer to leave this discussion in.

---

## [Editor Report · Decision Letter 1]

19 Aug 2020

The daytime feeding frequency affects appetite-regulating hormones, amino acids, physical activity, and respiratory quotient, but not energy expenditure, in adult cats fed regimens for 21 days

PONE-D-20-15734R1

Dear Dr. Shoveller,

We’re pleased to inform you that your manuscript has been judged scientifically suitable for publication and will be formally accepted for publication once it meets all outstanding technical requirements.

Kind regards,

Juan J Loor

Academic Editor

PLOS ONE
---

## [Editor Report · Acceptance letter]

1 Sep 2020

PONE-D-20-15734R1 

The daytime feeding frequency affects appetite-regulating hormones, amino acids, physical activity, and respiratory quotient, but not energy expenditure, in adult cats fed regimens for 21 days 

Dear Dr. Shoveller:

I'm pleased to inform you that your manuscript has been deemed suitable for publication in PLOS ONE. Congratulations! Your manuscript is now with our production department. 

Kind regards, 

on behalf of

Dr. Juan J Loor 

Academic Editor

PLOS ONE